# Evidence of Neuroinflammation and Blood–Brain Barrier Disruption in Women with Preeclampsia and Eclampsia

**DOI:** 10.3390/cells10113045

**Published:** 2021-11-05

**Authors:** Lina Bergman, Roxanne Hastie, Henrik Zetterberg, Kaj Blennow, Sonja Schell, Eduard Langenegger, Ashley Moodley, Susan Walker, Stephen Tong, Catherine Cluver

**Affiliations:** 1Department of Obstetrics and Gynecology, Stellenbosch University, Cape Town 7500, South Africa; sonjaschell@sun.ac.za (S.S.); langen@sun.ac.za (E.L.); ashleymoodley69@gmail.com (A.M.); cathycluver@hotmail.com (C.C.); 2Department of Women’s and Children’s Health, Uppsala University, 75309 Uppsala, Sweden; hastie.r@unimelb.edu.au; 3Department of Obstetrics and Gynecology, Institute of Clinical Sciences, Sahlgrenska Academy, University of Gothenburg, 41685 Gothenburg, Sweden; 4Translational Obstetrics Group, Department of Obstetrics and Gynecology, University of Melbourne, Heidelberg, VIC 3084, Australia; spwalker@unimelb.edu.au (S.W.); stong@unimelb.edu.au (S.T.); 5Mercy Perinatal, Mercy Hospital for Women, Heidelberg, VIC 3084, Australia; 6Institute of Neuroscience and Physiology, Department of Psychiatry and Neurochemistry, Sahlgrenska Academy, University of Gothenburg, 41345 Mölndal, Sweden; henrik.zetterberg@clinchem.gu.se (H.Z.); kaj.blennow@neuro.gu.se (K.B.); 7Clinical Neurochemistry Laboratory, Sahlgrenska University Hospital, 41385 Mölndal, Sweden; 8UCL Institute of Neurology, Department of Neurodegenerative Disease, Queen Square, London WC1N3BG, UK; 9UK Dementia Research Institute, London WC1E6BT, UK; 10Hong Kong Center for Neurodegenerative Diseases, Clear Water Bay, Kowloon, Hong Kong 999077, China

**Keywords:** eclampsia, neuroinflammation, blood–brain barrier, preeclampsia, cerebral edema

## Abstract

Cerebral complications in preeclampsia are leading causes of maternal mortality. Animal models suggest that an injured blood–brain barrier and neuroinflammation may be important but there is paucity of data from human studies. Therefore, we aimed to evaluate this in women with preeclampsia and eclampsia. We included women recruited to the South African Preeclampsia Obstetric Adverse Events (PROVE) biobank. Blood and cerebrospinal fluid (CSF) were collected around delivery. CSF was analyzed for neuroinflammatory markers interleukin 1β, interleukin 6, interleukin-8 and tumor necrosis factor alpha (TNF-alpha). The CSF to plasma albumin ratio was measured to assess blood–brain barrier function. Women with eclampsia (*n* = 4) showed increased CSF concentrations of all pro-inflammatory cytokines and TNF-alpha compared to women with normotensive pregnancies (*n* = 7) and also for interleukin-6 and TNF-alpha compared to women with preeclampsia (*n* = 4). Women with preeclampsia also showed increases in pro-inflammatory cytokines IL-6 and IL-8 but not TNF-alpha in the CSF compared to women with normotensive pregnancies. In particular, women with eclampsia but also women with preeclampsia showed an increase in the CSF to plasma albumin ratio compared to normotensive women. In conclusion, women with preeclampsia and eclampsia show evidence of neuroinflammation and an injured blood–brain barrier. These findings are seen in particular among women with eclampsia.

## 1. Introduction

Preeclampsia is a multisystem disorder that complicates 4–6% of all pregnancies [1,2]. The cerebral complications of preeclampsia, which include eclampsia, cerebral edema and stroke, are a leading cause of severe maternal morbidity and mortality [3,4]. Additionally, long-term neurological consequences of preeclampsia and its complications include an increased risk for white matter lesions, stroke, seizure disorders and vascular dementia later in life [5,6,7].

Magnesium sulfate is the drug of choice to prevent and treat eclampsia and has shown effects in reducing maternal mortality and morbidity [4,8]. The mechanism of action is not completely understood but animal studies suggest that magnesium sulfate may decrease neuroinflammation and potentially protect the functionality of the blood–brain barrier [9,10,11,12].

Animal models have demonstrated an impaired blood–brain barrier and increased neuroinflammation in preeclampsia. In these studies, animals with induced preeclampsia demonstrated an increased number of activated microglial cells and an increased cerebrospinal fluid (CSF) concentration of cytokines. In addition, the CSF/plasma albumin ratio was increased in preeclampsia as a measurement of blood–brain barrier impairment [10,12,13,14]. So far, there has been no evidence from pregnant women with preeclampsia to support this finding, with only one study having found increased levels of interleukins in CSF in preeclampsia, although other studies have not [15,16,17,18]. To our knowledge, no studies have assessed neuroinflammation and the blood–brain barrier in women with eclampsia.

We, therefore, evaluated whether there is evidence of blood–brain barrier disruption and neuroinflammation in women with eclampsia compared to those with preeclampsia and normotensive women.

## 2. Materials and Methods

### 2.1. Study Cohort

We included women who were recruited into the South African Preeclampsia Obstetric Adverse Events (PROVE) biobank and database that had a cesarean section delivery. The PROVE biobank is an ongoing collaborative project that facilitates research in the field of preeclampsia, with a focus on phenotyping severe disease. Women diagnosed with preeclampsia and normotensive controls are enrolled in the biobank at admission to Tygerberg Hospital (Stellenbosch University, Cape Town, South Africa). Biological samples and clinical data are collected at inclusion or delivery, during the hospital stay and postpartum [19]. Tygerberg Hospital is the largest referral hospital in the Western Cape Provence of South Africa. In 2018 there were 32,422 deliveries in the referral area, of which 8067 were considered high risk and delivered at Tygerberg. PROVE biobank includes the majority of women with eclampsia presenting at the hospital, where around 50 women with eclampsia are recruited yearly [19].

For this study, we included women with singleton pregnancies. Exclusion criteria were known neurological and cardiac diseases. For normotensive women, additional exclusion criteria were chronic hypertension and diabetes mellitus. Preeclampsia and eclampsia were defined according to the American College of Obstetricians and Gynecologists (ACOG) 2020 Practice Bulletin [20]. A woman was considered normotensive if she had no documented systolic blood pressure greater than or equal to 140 mmHg or a diastolic blood pressure greater than or equal to 90 mmHg during her pregnancy until discharge postpartum. All women with eclampsia were recruited after the first generalized seizure and all women with preeclampsia were recruited after diagnosis. All women were recruited before delivery.

Baseline data were obtained by interview and extraction from medical records. All data were entered and stored using REDCap (Research Electronic Data Capture) tools hosted at Stellenbosch University [21]. Electronic data were double-checked for accuracy and cross-referenced with original data collection forms collected by PROVE research midwives.

### 2.2. Sample Collection

Blood was drawn on the day before delivery (normotensive control = 1), at delivery (normotensive controls = 2, preeclampsia = 1, eclampsia = 3), one day postpartum (normotensive controls = 3, eclampsia = 1), two days postpartum (normotensive control = 1, preeclampsia = 1), three days postpartum (preeclampsia = 1) and four days postpartum (preeclampsia = 1). CSF was collected at the time of spinal anesthesia at delivery. CSF was analyzed for concentrations of neuroinflammatory markers interleukin 1 (IL-1β), interleukin 6 (IL-6), interleukin-8 (IL-8) and tumor necrosis factor alpha (TNF-alpha). The CSF/plasma albumin ratio was measured to assess blood–brain barrier integrity.

Blood samples were collected in a 9 mL ethylenediaminetetraacetic acid (EDTA) tube at time of inclusion in the study. CSF samples were collected at cesarean section delivery at the time of spinal anesthesia. All samples were centrifuged for 10 min at 2800× *g* and plasma and CSF were aliquoted in 2 mL cryotubes and subsequently frozen at −80 °C until the analysis.

### 2.3. Biomarker Assays

All CSF measurements and plasma/serum concentrations were assessed in one round of experiments using a single batch of reagents for each assay by laboratory technicians who were blinded to clinical data and groups.

CSF and plasma albumin concentrations were measured by immunonephelometry on a Beckman Immage Immunohistochemistry system (Beckman Instruments, Beckman Coulter, Brea, CA, USA). The albumin ratio was calculated as CSF albumin (mg/L)/serum albumin (g/L) and was used as a measure of the blood–brain barrier integrity [22]. CSF concentrations of IL-1β, IL-6, IL-8 and TNF-α were measured using the Meso Scale Discovery 4-plex Proinflammatory Panel II according to instructions from the kit manufacturer (Meso Scale Discovery, Rockville, MD). The individual assays in the multiplexed inflammation marker panel (the Meso Scale Discovery 4-plex Proinflammatory Panel II) have been validated for research use in human CSF, as previously described (https://www.mesoscale.com/~/media/files/scientific%20poster/measuring-low-concentrations-cytokines-csf-assays-aai-2017-msd.pdf?la=en, accessed on 5 November 2021). The measurement ranges were 0.284–1166 pg/mL for IL-1β, 0.324–1326 pg/mL for IL-6, 0.28–1150 pg/mL for IL-8 and 0.165–676 pg/mL for TNF-α. All measurements were performed in one round of experiments using one batch of reagents. The intra-assay coefficient of variation for internal control samples within the linear range was <10%.

### 2.4. Statistics

Demographic and clinical characteristics are presented as means with standard deviations (SD) and percentages. Albumin quotient and neuroinflammatory markers are presented as medians with ranges and compared between groups as fold-changes with 95% confidence intervals.

Due to the small sample sizes, *p*-values for differences in concentrations of inflammatory markers in CSF and albumin ratios were calculated using an exact permutation test for the mean difference on log-transformed variables. Corresponding confidence intervals were calculated by test inversion.

In all hypothesis tests, a two-sided *p*-value of less than 0.05 was considered statistically significant. Data and statistical analyses were performed using SPSS version 26.0 (SPSS; PASW statistics, IBM corp) and SAS version 9.4 (SAS Institute INC, Cary, NC, USA).

### 2.5. Ethical Permission and Registration Details

Ethical approval was obtained (protocol number N18/03/034, Federal Wide assurance number 00001372, Institutional Review Board number IRB0005239) and all included participants signed informed consent before being enrolled in the Preeclampsia Obstetric Adverse Events (PROVE) biobank. The biobank is registered at ISRCTN with registration number ISRCTN10623443 and the protocol has been published previously [19].

## 3. Results

We included women enrolled in the PROVE biobank between April 2018 and March 2020. During this time, 233 women with biological samples were included in the PROVE Biobank and 15 of these had CSF collected. Seven women were normotensive, four had preeclampsia and four had eclampsia (Figure 1).

Maternal characteristics and pregnancy outcomes are presented in Table 1.

Compared to normotensive women, those who had eclampsia were younger and more often nulliparous. Women with preeclampsia or eclampsia delivered at an earlier gestation and had lower birthweight infants. All women with preeclampsia and eclampsia gave birth by emergency cesarean section. Women in the normotensive group all had elective cesarean sections. One woman with eclampsia and one with preeclampsia had severe hypertension. There were no other neurological complications such as stroke and no women were admitted to the general intensive care unit, had a low Glasgow coma scale or needed intubation. None experienced pulmonary edema or hemolysis, elevated liver enzyme and low platelet (HELLP) syndrome. All women with eclampsia and preeclampsia were treated with magnesium sulfate until after delivery. All women in the study received cephalosporin antibiotics at the time of cesarean section.

### 3.1. Neuroinflammatory Markers

When compared to normotensive pregnancies, women with eclampsia showed evidence of increased neuroinflammation. Those with eclampsia showed an 18.0-fold increase in CSF concentrations of IL-6 (95% CI 3.17–121.15, *p* = 0.006), a 7.8-fold increase in IL-8 (95% CI 3.58–18.75, *p* = 0.006) and a 3.7-fold increase in TNF-alpha (95% CI 1.53–7.70, *p* = 0.006). There was also evidence of increased neuroinflammation among those who had eclampsia compared to women with preeclampsia: a 4.6-fold increase in IL-8 (95% CI 1.66–40.92, *p* = 0.029), a 3.2-fold increase in TNF-alpha (95% CI 1.10–22.53, *p* = 0.029) and a non-significant 8.5-fold increase in IL-6 (95% CI 0.86–521.37, *p* = 0.057).

When women with preeclampsia were compared to normotensive women there was a 2.1-fold increase in IL-6 (95% CI 1.18–3.85, *p* = 0.024) and a 1.7-fold increase in IL-8 (95% CI 1.03–2.79, *p* = 0.036) (Figure 2 and Table 2 and Table 3).

IL-1β CSF concentrations were also measured, although concentrations were generally below the limit of detection. One woman with eclampsia showed concentrations of the inflammatory markers 10 times higher than the others in the eclampsia group. Excluding the results of this participant did not significantly change the findings.

### 3.2. Albumin Quotient

There was also evidence of significant disruption of blood–brain barrier integrity among those who had eclampsia and even preeclampsia. Women with eclampsia showed a 3.8-fold increase in the CSF/plasma albumin ratio (95% CI 2.33–6.48, *p* = 0.006) when compared to normotensive women. Women with preeclampsia showed a 1.9-fold increase in the CSF/plasma albumin ratio (95% CI 1.01–3.65, *p* = 0.048) compared to normotensive women (Figure 2 and Table 3). When restricting the analyses by matching eclampsia and normotensive controls for day of plasma sampling (*n* = 4 in each group), the plasma/albumin ratio in women with eclampsia remained increased compared to controls (median, range, 10.4, 8.2–23.2 vs. 3.3, 2.8–4.1, *p* = 0.029).

## 4. Discussion

### 4.1. Principal Findings

Inflammatory markers (IL-6, IL-8 and TNF-alpha) were increased in the CSF of women with eclampsia, compared to those with preeclampsia and women with normotensive pregnancies. Furthermore, there was evidence of injury to the blood–brain barrier with an increased CSF/plasma albumin ratio among women with eclampsia compared to normotensive women. There was also evidence of injury among those with preeclampsia, though it was to a lesser extent. This study provides evidence of neuroinflammation and blood–brain barrier injury in women with eclampsia and preeclampsia.

### 4.2. Results in Context

To the best of our knowledge, there are no previous studies reporting on CSF findings among women with eclampsia. There have been a few studies that have measured protein content in CSF from women with preeclampsia. One study that used SOMAscan proteomics with CSF samples from 13 women with preeclampsia and 14 women with normotensive pregnancies demonstrated increased levels of IL-8 and IL-9 in CSF from women with preeclampsia, although only related levels were reported and no information was given about absolute concentrations [17]. Another study analyzed CSF samples using liquid chromatography tandem mass spectrometry proteomics in 43 women with severe preeclampsia and 55 normotensive women [18]. Of the 457 proteins analyzed, 25 were different between those with preeclampsia and normotensive controls. There were eight proteins that were more abundant in preeclampsia and 18 that were less abundant. The most significantly abundant protein was protein alpha-1-microglobulin/bikunin precursor (AMBP), a precursor of alpha-1-microglobulin (A1M) that is a heme scavenger and potentially protective against oxidative stress caused by free hemoglobin (Hb). The other proteins that were more or less abundant in preeclampsia included acute phase proteins and proteins related to angiogenesis. Their analyses did not include interleukins or TNF-alpha [18]. The CSF/plasma albumin ratio was not assessed in these studies [17,18].

The CSF/plasma albumin ratio is an expression of the integrity of the blood–brain barrier, where the plasma-derived protein albumin should be found in very low concentrations in the CSF in normal conditions, resulting in a low CSF/plasma albumin ratio. In one study that had previously investigated the CSF/plasma albumin ratio, 15 women with preeclampsia and 15 women with normotensive pregnancies were included. There were no women with eclampsia in this study [15]. One woman in the preeclampsia group had HELLP syndrome but none had eclampsia or other severe end organ complications. In the normotensive group, the CSF/plasma albumin ratio was 2.73, similar to our findings of 3.1. Women with preeclampsia had a median CSF/plasma albumin ratio of 3.02 [15]. In our study, the median CSF/plasma albumin ratio in women with preeclampsia was 5.2. The only study we could find that investigated both neuroinflammation and blood–brain barrier integrity in preeclampsia assessed the concentrations of C5a, C5b-9, TNF-alpha and IL-6 in CSF and the CSF/plasma albumin ratio in preeclampsia cases with or without severe features (*n* = 16), compared to hypertensive disease (*n* = 16) without proteinuria and normotensive controls (*n* = 16) [16]. In this study, severe disease was mostly due to severe hypertension, with only one case of HELLP syndrome and one case of pulmonary edema. No difference was found between groups for any of the inflammatory markers or the CSF/plasma albumin ratio [16]. In this study, the overall CSF concentrations of IL-6 were higher than in our study but not different between preeclampsia and normotensive women. The CSF concentrations of TNF-alpha and the CSF/plasma albumin ratio in normotensive women were similar to our study. Interestingly, women with preeclampsia had a lower CSF/plasma albumin ratio of 3.5 (IQR 2.9–5.1) compared to 5.21 (range 2.83–14.79) in our study. These differences may be due to different assays with variable intra- and inter-coefficients or differences in the disease phenotype. The previous study recruited women from a high-income setting where preeclampsia may be identified earlier, potentially resulting in a shorter time of exposure to harmful molecules that could injure the blood–brain barrier compared to our study where women often presented late.

Perhaps owing to the difficulty of studying blood–brain barrier integrity and pathophysiology in women with preeclampsia, several animal models have been established. The reduced uterine perfusion pressure (RUPP) model in rats or mice is one of the most commonly used. Studies have reported increased permeability of the blood–brain barrier to the larger molecule dextran [14] and others only to the smaller molecule sodium fluorescein [10]. RUPP rats also had increased cerebral water content two months postpartum and this was thought to be a good proxy of persistent cerebral edema [23]. Neuroinflammatory activity measured using immunohistochemistry in this RUPP model also showed increased neuroinflammatory activity, which was interestingly reversed by magnesium sulfate treatment [10].

Our group previously reported on in vitro studies of the human brain endothelium, exposed to plasma from women with preeclampsia [24]. We demonstrated increased permeability and a lower trans endothelial electrical resistance over the in vitro blood–brain barrier after exposure to plasma from women with preeclampsia compared to normotensive women, implicating that plasma from women with preeclampsia induces injury to the blood–brain barrier [24], supporting the findings from animal studies.

This study supports data from preclinical studies and suggests that neuroinflammation and blood–brain barrier injury may be important in the pathophysiology of eclampsia and preeclampsia.

### 4.3. Clinical Implications

In this study, women with eclampsia had a median albumin quotient of 10.4 with a range of 8.23–23.17. These are all above the reference value of <6.5 for corresponding age and gender [25,26,27]. This suggests that eclampsia represents a pathological state severely affecting blood–brain barrier integrity.

The novel findings of blood–brain barrier injury and increased neuroinflammation in preeclampsia and eclampsia in the acute phase may be important for short-term complications and the risk for long-term neurological consequences of preeclampsia [5,6,7,28].

The presence of an injured blood–brain barrier and neuroinflammation in other medical conditions such as traumatic brain injury have been suggested as underlying factors of long-term neurological outcome, such as cognitive dysfunction and chronic traumatic encephalopathy [29,30]. There are similar hypotheses in the field of neurodegenerative disease with an initial insult to the blood–brain barrier and subsequent neuroinflammation as an underlying trigger to the development of later neurodegenerative disease [31].

Treatments that restore or protect blood–brain barrier integrity and decrease neuroinflammation could be important in preventing and treating eclampsia and may possibly protect women from long-term cerebral adverse outcomes associated with preeclampsia, such as cognitive decline, epilepsy and dementia [5,6,32].

### 4.4. Research Implications

Larger studies should be done to validate our findings and further evaluate the effects of magnesium sulfate and other neuroprotective drugs on the degree of neuroinflammation and blood–brain barrier integrity in a clinical cohort of women with preeclampsia and eclampsia. If confirmed, future research should focus on neuroprotective treatment for women with preeclampsia and neurological complications to decrease neuroinflammation and restore blood–brain barrier integrity with short- and long-term effects.

### 4.5. Strengths and Limitations

We have collected unique CSF samples in a cohort of women with eclampsia. It is a difficult task to obtain CSF samples in eclampsia, as these women are often delivered in an emergency situation after hours and they often require general anesthesia due to other complications including HELLP syndrome, which precludes CSF sampling. Collecting these samples was only possible due to the high incidence of eclampsia at our hospital and the infrastructure of the PROVE biobank. Even so, we were only able to collect CSF samples from four women with eclampsia with corresponding controls with a total of 233 women in the biobank, which highlights the difficulty of obtaining CSF samples at delivery in eclampsia. A limitation is that not all plasma samples were drawn on the day of delivery, inducing potential bias in the interpretation of the CSF/plasma albumin ratio. However, when restricting the analyses by matching eclampsia and normotensive controls for day of plasma sampling, the plasma/albumin ratio in women with eclampsia remained increased compared to controls. Another limitation is that our sample size was small. Despite this, we were still able to demonstrate significant differences in neuroinflammation and blood–brain barrier integrity between the groups, which implies that the differences are very significant. If a larger sample size could be obtained in future studies, it would be of value to perform adjusted models correcting for potential confounders such as BMI, maternal age and parity.

## 5. Conclusions

Women with eclampsia show evidence of neuroinflammation and an injured blood–brain barrier. This provides important knowledge in understanding the underlying pathophysiological mechanisms in eclampsia and may help direct therapeutic interventions to prevent and treat neurological complications of preeclampsia (such as anti-inflammatory agents after an eclamptic seizure).

## Figures and Tables

**Figure 1 cells-10-03045-f001:**
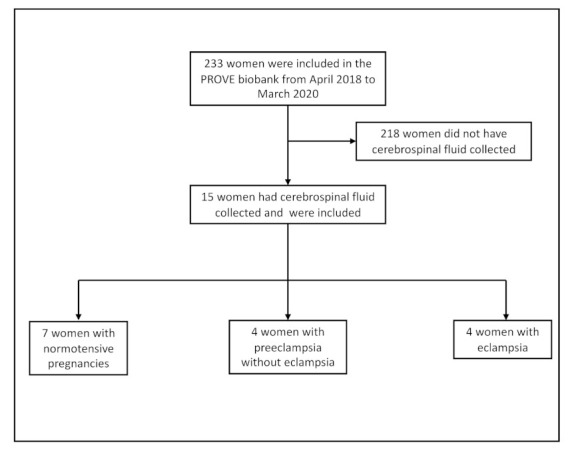
Background characteristics.

**Figure 2 cells-10-03045-f002:**
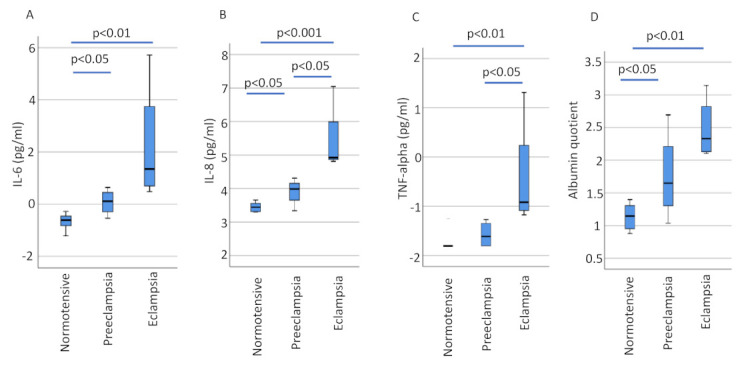
Log-transformed concentrations of neuroinflammatory markers and albumin quotient in CSF samples. Boxplot showing log-transformed CSF concentrations (IL-6, IL-8 and TNF-alpha) and log-transformed CSF/plasma albumin ratios as medians with interquartile ranges for IL-6 (**A**), IL-8 (**B**), TNF alpha (**C**) and albumin (**D**). Normotensive pregnancy (*n* = 7), preeclampsia (*n* = 4) and eclampsia (*n* = 4). IL-6, interleukin 6; IL-8, interleukin 8; TNF-alpha, tumor necrosis factor alpha.

**Table 1 cells-10-03045-t001:** Background characteristics of the population.

	Normotensiv Pregnancy	Preeclampsia	Eclampsia
n	7	4	4
At baseline			
Maternal age (years)	30.9 (5.0)	27.3 (6.4)	18.3 (4.2)
Nulliparous (%)	1 (14)	2 (50)	3 (75)
HIV positive(%)	1 (14)	1 (25)	0 (0)
Smoking (%)	1 (14)	1 (25)	0 (0)
Diabetes (%)	0 (0)	0 (0)	0 (0)
Chronic hypertension (%)	0 (0)	2 (50)	0 (0)
BMI (kg/m^2^)	31.6 (5.7)	19.8 (2.3) *	24.6 (4.8)
**After inclusion**			
GA at delivery (weeks)	38.4 (0.8)	33.3 (3.8)	34.3 (3.6)
Magnesium sulfate (%)	0 (0)	3 (75)	4 (100)
Hours before CSF sample	N/A	3.5 (1.2–25.5)	23 (12.5–30)
**Mode of delivery (%)**			
Vaginal delivery	0 (0)	0 (0)	0 (0)
Elective CS	7 (100)	0 (0)	0 (0)
Emergency CS	0 (0)	4 (100)	4 (100)
Liveborn (%)	7 (100)	4 (100)	4 (100)
Birthweight (g)	3279.3 (242.7)	1713.8 (693.9)	2235.0 (1087.3)
**Maternal complications**			
Recurrent eclampsia	0 (0)	0 (0)	3 (75)
Severe hypertension	0 (0)	1 (25)	1 (25)

Abbreviations: HIV = human immune deficiency virus. * Missing data on one participant. Severe hypertension was defined as a systolic blood pressure of >160 mm Hg and/or a diastolic blood pressure of >110 mm Hg despite intravenous treatment. No women in this study reported alcohol or methamphetamine use during pregnancy.

**Table 2 cells-10-03045-t002:** Cerebrospinal fluid concentrations of neuroinflammatory markers and CSF/plasma albumin ratios.

	Normotensive	Preeclampsia	Eclampsia
n	7	4	4
IL-6 (pg/mL)	0.54 (0.30–0.76)	1.13 (0.59–1.90)	4.23 (1.63–305.00)
IL-8 (pg/mL	31.36 (16.66–38.75)	53.88 (28.10–74.45)	137.81 (123.43–1150.00)
TNF-alpha (pg/mL)	0.17 (0.17–0.30)	0.20 (0.17–0.28)	0.40 (0.31–3.72)
CSF/plasma albumin ratio	3.15 (2.41–4.05)	5.21 (2.83–14.79)	10.42 (8.23–23.17)

CSF, cerebrospinal fluid; IL-6, interleukin 6; IL-8, interleukin 8; n, number’ TNF-alpha, tumor necrosis factor-alpha. Numbers are presented as medians with ranges.

**Table 3 cells-10-03045-t003:** Fold changes of cerebrospinal fluid concentrations of cerebral biomarkers, neuroinflammatory markers and CSF/plasma albumin ratios.

Biomarker	Preeclampsia vs. Normotensive	Eclampsia vs. Normotensive	Eclampsia vs. Preeclampsia
IL-6 (pg/mL)	2.11 (1.18–3.85) *p* = 0.024	18.03 (3.17–121.15) *p* = 0.006	8.53 (0.86–521.37) *p* = 0.057
IL-8 (pg/mL)	1.69 (1.03–2.79) *p* = 0.036	7.76 (3.58–18.75) *p* = 0.006	4.59 (1.66–40.92) *p* = 0.029
TNF-alpha (pg/mL)	1.16 (0.75–1.58) *p* = 0.56	3.65 (1.53–7.70) *p* = 0.006	3.16 (1.10–22.53) *p* = 0.029
CSF/plasma albumin ratio	1.86 (1.01–3.65) *p* = 0.048	3.83 (2.33–6.48) *p* = 0.006	2.05 (0.56–8.19) *p* = 0.14

Data are presented as fold changes with 95% confidence intervals; *p*-values were calculated using exact permutation test for the mean difference on log-transformed variables. Corresponding confidence intervals were calculated by test inversion. Abbreviations: CSF, cerebrospinal fluid; IL-6, interleukin 6; IL-8, interleukin 8; TNF-alpha, tumor necrosis factor alpha.

## Data Availability

Anonymized data not published within this article will be made available on request from any qualified investigator after approval, as described in the PROVE protocol [19].

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
