# Peer review of "Evidence of Neuroinflammation and Blood–Brain Barrier Disruption in Women with Preeclampsia and Eclampsia"

_cells, 2021, doi:10.3390/cells10113045_

Round 1

Reviewer 1 Report

This study evaluated blood-brain barrier disruption and neuroinflammation in women with eclampsia, compared to those with preeclampsia and normotensive women. This study showed evidence of neuroinflammation and an injured blood-brain barrier using plasma and CSF. This is the first study to evaluate neuroinflammation and a blood-brain barrier in eclampsia however number of study objects is small.  

This is an interesting study however there is a major pitfall.

  1. Blood-brain barrier was confirmed by CSF albumin to plasma albumin and while CSF albumin was checked before delivery, plasma albumin was checked at different period such as postpartum or antepartum. There is significant hemodynamic physiologic change between postpartum and antepartum period therefore plasma albumin should be checked at a similar time when CSF albumin was checked and compared. This limitation is stated in page 10, first paragraph and ‘data not shown’ should be stated in the table or elsewhere in this manuscript.
  2. The exact time of magnesium sulphate use should be stated. Magnesium sulphate might influence inflammatory factors and albumin.

Author Response

Reviewer #1

This study evaluated blood-brain barrier disruption and neuroinflammation in women with eclampsia, compared to those with preeclampsia and normotensive women. This study showed evidence of neuroinflammation and an injured blood-brain barrier using plasma and CSF. This is the first study to evaluate neuroinflammation and a blood-brain barrier in eclampsia however number of study objects is small.  

This is an interesting study however there is a major pitfall.

  • Blood-brain barrier was confirmed by CSF albumin to plasma albumin and while CSF albumin was checked before delivery, plasma albumin was checked at different period such as postpartum or antepartum. There is significant hemodynamic physiologic change between postpartum and antepartum period therefore plasma albumin should be checked at a similar time when CSF albumin was checked and compared. This limitation is stated in page 10, first paragraph and ‘data not shown’ should be stated in the table or elsewhere in this manuscript.

We thank the reviewer for this important aspect and as you can see, we have also contemplated on the importance of this. For this study, due to the acute situations when women are delivered with preeclampsia and eclampsia, we were not always able to obtain the blood samples at the same moment as the cerebrospinal fluid. We have now added information on this in the result section and it now reads;

“When restricting the analyses by matching eclampsia and normotensive controls for day of plasma sampling (n=4 in each group), plasma-albumin ratio in women with eclampsia remained increased compared to controls (median, range, 10.4; 8.2-23.2 vs 3.3, 2.8-4.1, p=0.029).”

Page 8, lines 220-223

In addition, we removed “data not shown” on page 10 in the discussion.

  • The exact time of magnesium sulphate use should be stated. Magnesium sulphate might influence inflammatory factors and albumin.

This is also an important comment. To our knowledge, it is not yet known how magnesium sulphate affects the content of neuroinflammatory markers in the cerebrospinal fluid in humans whereas it has been shown to reduce neuroinflammatory activity through reduced expression of active neuroglial cells in animal models of preeclampsia. In this study, all women with eclampsia and preeclampsia had received magnesium sulphate before delivery whereas none in the normotensive groups did. If anything, this would have reduced the difference in neuroinflammatory markers between the group, should magnesium sulphate had lowered the neuroinflammatory activity in women with preeclampsia and eclampsia. In our study, magnesium sulfate was administered between 30-1 hours before delivery. These details have now been added to Table 1.

Reviewer 2 Report

The study aims to evaluate if there is evidence of injured blood-brain barrier and neuroinflammation in women with preeclampsia and eclampsia. It represents an innovative study in literature. Although the authors were able to demonstrate significant differences in neuroinflammation as well as in blood-brain barrier integrity between groups, the main limitation of the study is the small sample size of the groups studied. The study was developed with blood and cerebrospinal fluid (CSF) of four women with preeclampsia, four with eclampsia, and seven normotensive pregnant women recruited as controls. All the samples were obtained from PROVE biobank.  

There are several major issues that should be addressed:

  1. In the introduction more information about references 15-18 may be interesting for a better understanding of neuroinflammation (lines 53-55).

Material and methods:

  1. The authors did not calculate the sample size to consider the difference between the studied groups. As Tygerberg Hospital is the largest referral hospital in the Western Cape Provence of South Africa, it is supposed to receive a significant number of preeclamptic and eclamptic patients, since the authors informed the hospital receives around 50 women with eclampsia yearly. The study included women enrolled in the PROVE biobank between April 2018 and March 2020.
  2. The exclusion criteria for normotensive women were chronic hypertension and diabetes. Did other pre-existing medical conditions such as acute infectious diseases, autoimmune, renal, and hepatic diseases were also excluded? Table 1 shows patients HIV positive enrolled in normotensive and preeclamptic women groups.
  3. CSF was collected at the time of spinal anesthesia, at delivery. Why the blood collection wasn’t done at the same time of CSF?

  1. Results:

Table 1. Why there is no statistical analysis of the results? The authors didn’t mention if there are statistical differences among the groups in relation to maternal characteristics and pregnancy outcomes. No representative symbol of the differences was shown. 

Ex: Are there significant differences between groups in relation to maternal age, BMI, and birth weight?

Line 152: women with preeclampsia or eclampsia delivered at an earlier gestation and had lower birth weight infants. Are these differences significant compared to normotensive pregnancy? It should be asked whether the authors were limited to describe the results without showing statistical comparison, due to the small sample size?

In figure 2 and table 3, the results of inflammatory cytokines in the CSF and Albumin quotient show statistical differences among the groups.

  1. Discussion:

Although the study provides evidence of neuroinflammation and blood-barrier injury in women with eclampsia and preeclampsia, the small sample size studied must be considered as a limitation of the study. 

The authors mentioned that pregnant women with preeclampsia and eclampsia were submitted to treatment with magnesium sulfate. The literature shows by experimental models of preeclampsia and eclampsia and also in patients that magnesium sulfate has neuroprotective and anti-inflammatory activities. Thus, future studies must be carried out comparing the parameters of the present study in women with eclampsia submitted to treatment with magnesium sulfate, evaluated in different time points: before and after treatment with magnesium sulfate in a longitudinal study.

Author Response

Reviewer #2

The study aims to evaluate if there is evidence of injured blood-brain barrier and neuroinflammation in women with preeclampsia and eclampsia. It represents an innovative study in literature. Although the authors were able to demonstrate significant differences in neuroinflammation as well as in blood-brain barrier integrity between groups, the main limitation of the study is the small sample size of the groups studied. The study was developed with blood and cerebrospinal fluid (CSF) of four women with preeclampsia, four with eclampsia, and seven normotensive pregnant women recruited as controls. All the samples were obtained from PROVE biobank.  

We thank the reviewer for this positive feedback and we agree that the sample size is indeed small. Though this is due to the fact that it is challenging to obtain cerebrospinal fluid from women with eclampsia. Eclampsia is one of the most feared complications of preeclampsia and delivery is often initiated promptly, often after office hours and in many cases, the extra sample for the study is forgotten. In addition, in this setting, many of the women are anesthetized with generalized anesthesia due to causes such as low platelets or focal neurological symptoms, preluding obtaining cerebrospinal fluid at time of spinal anesthesia. Even in a center such as Tygerberg Hospital that carries a high burden of eclampsia, we only managed to obtain cerebrospinal fluid samples of four women with eclampsia over the course of two years.

There are several major issues that should be addressed:

  • In the introduction more information about references 15-18 may be interesting for a better understanding of neuroinflammation (lines 53-55).

We thank the reviewer for this comment and we have now added more information about neuroinflammation in general and it now reads:

“In these studies, animals with induced preeclampsia demonstrated increased number of activated microglial cells and an increased cerebrospinal fluid (CSF) concentration of cytokines. In addition, the CSF-plasma albumin ratio was increased in preeclampsia as a measurement of blood-brain barrier impairment.”

Page 2, lines 55-59

Material and methods:

  • The authors did not calculate the sample size to consider the difference between the studied groups. As Tygerberg Hospital is the largest referral hospital in the Western Cape Provence of South Africa, it is supposed to receive a significant number of preeclamptic and eclamptic patients, since the authors informed the hospital receives around 50 women with eclampsia yearly. The study included women enrolled in the PROVE biobank between April 2018 and March 2020.

Thank you for raising this point. We refer to our reply above regarding the difficulty to obtain cerebrospinal fluid samples in women with eclampsia. Since the sample is small and there have been no previous studies on neuroinflammatory markers in eclampsia, this has to be regarded as a pilot study. We can’t see how a post-hoc power analysis would have made any difference to the interpretation of the results. Though, if the reviewer wish, we could calculate a post-hoc power analyses from animal data.

  • The exclusion criteria for normotensive women were chronic hypertension and diabetes. Did other pre-existing medical conditions such as acute infectious diseases, autoimmune, renal, and hepatic diseases were also excluded? Table 1 shows patients HIV positive enrolled in normotensive and preeclamptic women groups.

In this study, two of the normotensive women had iron deficiency anemia and one was HIV positive. There were no women with other acute infectious diseases, autoimmune, renal or hepatic disease. All collected variables on current and previous disease can be found in the published protocol.(1)

  • CSF was collected at the time of spinal anesthesia, at delivery. Why the blood collection wasn’t done at the same time of CSF?

Thank you for this question and please see the reply to Reviewer 1, first question.

Results:

  • Table 1. Why there is no statistical analysis of the results? The authors didn’t mention if there are statistical differences among the groups in relation to maternal characteristics and pregnancy outcomes. No representative symbol of the differences was shown. Ex: Are there significant differences between groups in relation to maternal age, BMI, and birth weight?

We generally feel there is not right to include statistical inference calculations in the background table (Table 1) since the study is not designed to find differences in background data. In small samples sizes like this one, there might be differences in mean/median values even if it is not statistically proven. On the contrary, in large epidemiological studies, there will be many statistically significant differences in background data even though the true differences are small, due to very large sample sizes. Thus, we would like to leave it to the reader to decide if the differences between groups regarding background variables are of clinical value. If the reviewer feels strongly about including p-values in the Table 1, we could add that. 

  • Line 152: women with preeclampsia or eclampsia delivered at an earlier gestation and had lower birth weight infants. Are these differences significant compared to normotensive pregnancy? It should be asked whether the authors were limited to describe the results without showing statistical comparison, due to the small sample size?

We thank the reviewer for this comment and we refer to the reply above. The reviewer is right in the fact that the sample size is too small to include covariates in an adjusted analysis, such as an ANCOVA. If these results are repeated in further studies and another research group manages to collect a larger sample size, it would be interesting to look into possible confounders and correct for these.

  • In figure 2 and table 3, the results of inflammatory cytokines in the CSF and Albumin quotient show statistical differences among the groups.

Yes, despite a small sample size there were statistically significant differences in the CSF concentrations of cytokines and also a difference in the CSF/plasma albumin ratio, in particular in women with eclampsia.

Discussion:

  • Although the study provides evidence of neuroinflammation and blood-barrier injury in women with eclampsia and preeclampsia, the small sample size studied must be considered as a limitation of the study. 

We agree with the reviewer that this is indeed a limitation but we still argue the samples are unique and no one has never been able to obtain this kind of samples before to the best of our knowledge. Therefore, we think the data is important and contributes to important knowledge in the field, also supporting the findings from animal models

  • The authors mentioned that pregnant women with preeclampsia and eclampsia were submitted to treatment with magnesium sulfate. The literature shows by experimental models of preeclampsia and eclampsia and also in patients that magnesium sulfate has neuroprotective and anti-inflammatory activities. Thus, future studies must be carried out comparing the parameters of the present study in women with eclampsia submitted to treatment with magnesium sulfate, evaluated in different time points: before and after treatment with magnesium sulfate in a longitudinal study.

Indeed, magnesium sulfate has been evaluated in the context of preeclampsia, mostly in animal and in vitro studies, and has been shown to act neuroprotective probably through effects on the NMDA receptor, through anti-inflammatory actions and/or perhaps also by acting protective on the blood-brain barrier. Though there is still a lack of knowledge regarding the mechanism of action in women with eclampsia and preeclampsia. It would indeed be interesting to evaluate the neuroinflammatory activity before, during and after treatment with magnesium sulfate. Though this would be an impossible task for two reasons. The first is that the drug should be administered to a woman with eclampsia as soon as she starts fitting and should not be withheld in order to obtain biological samples. Secondly, it would be ethically questionable to obtain cerebrospinal fluid samples at three different time points without a clinical indication. In this population, we obtained cerebrospinal fluid samples at the time of spinal anesthesia administered before the caesarian section to avoid an unnecessary intervention.

Reviewer 3 Report

In the manuscript, you presented an interesting topic in order to expand the knowledge about the degree of damage to the blood-brain barrier in hypertensive disorders during pregnancy. As you can see, the proportion of women in labour where CSF withdrawal is indicated is very small and therefore it cannot be expected that the determination of inflammatory markers and albumin ratios will be considered on a routine basis, and even in emergencies, CSF is unlikely. Importantly, however, you were able to obtain these samples from the biobank in at least a small number of cases and thus show what the values of neuroinflammatory markers are in the CSF and what the albumin ratio is.

Determination of markers in CSF: ordinary reagents are optimized for serum (plasma) determination. The concentrations of neuroinflammatory markers are different like in serum. Please add quality control data for CSF.

As you can see for yourself, the study is done on a small number of cases. Given that you have a protocol in place, I advise you to continue sampling to get more relevant results over time. 

Author Response

Reviewer #3

In the manuscript, you presented an interesting topic in order to expand the knowledge about the degree of damage to the blood-brain barrier in hypertensive disorders during pregnancy. As you can see, the proportion of women in labour where CSF withdrawal is indicated is very small and therefore it cannot be expected that the determination of inflammatory markers and albumin ratios will be considered on a routine basis, and even in emergencies, CSF is unlikely. Importantly, however, you were able to obtain these samples from the biobank in at least a small number of cases and thus show what the values of neuroinflammatory markers are in the CSF and what the albumin ratio is.

We thank the reviewer for this point. We agree that this is not meant to be a clinical routine test in eclampsia but the value of our findings is rather interesting from a pathophysiological point of view.

  • Determination of markers in CSF: ordinary reagents are optimized for serum (plasma) determination. The concentrations of neuroinflammatory markers are different like in serum. Please add quality control data for CSF.

We thank the reviewer for raising this point. To clarify, we’ve now added below text in the methods section:

“The individual assays in the multiplexed inflammation marker panel (the Meso Scale Discovery 4-plex Proinflammatory Panel II) have been validated for research use in human CSF, as previously described (https://www.mesoscale.com/~/media/files/scientific%20poster/measuring-low-concentrations-cytokines-csf-assays-aai-2017-msd.pdf?la=en). The measurement range was 0.284-1166 pg/mL for IL-1β, 0.324-1326 pg/mL for IL-6, 0.28-1150 pg/mL for IL-8, and 0.165-676 pg/mL for TNF-α. All measurements were performed in one round of experiments using one batch of reagents. Intra-assay coefficient of variation for internal control samples within the linear range were <10%.”

Page 3-4, lines 121-132

  • As you can see for yourself, the study is done on a small number of cases. Given that you have a protocol in place, I advise you to continue sampling to get more relevant results over time.

We thank the reviewer for raising this important point, also stressed by us and the other two reviewers. We are aiming to continue all biological sampling within the study and we can hopefully within a few years confirm these first unique data with follow-up analyses.

Reviewer 4 Report

The article is well designed, with important discoveries.

Make it clear on the test whether blood and CSF were collected BEFORE or after magnesium.

Also, discuss whether BMI between groups might underestimate the difference on biomarkers.

Do you think that the protein total in CSF from PE and E may be higher than normotensive. Do you measured the protein total level?

Author Response

Reviewer #4

The article is well designed, with important discoveries.

  • Make it clear on the test whether blood and CSF were collected BEFORE or after magnesium.

We have now clarified when the magnesium sulfate was administered in Table 1 according to reviewer’s request. All blood-samples for women with preeclampsia and eclampsia were drawn at the day of delivery up to a few days post delivery (stated in the methods section, lines 98-102) which is after administration of magnesium sulfate for all women.

  • Also, discuss whether BMI between groups might underestimate the difference on biomarkers.

BMI was indeed different in all groups. With increased BMI, plasma volume also increases. Though, women with an increased BMI do not seem to have a decreased concentration of albumin in plasma. If this was the case, it would rather have reduced the differences in plasma-albumin quotient between groups since the highest BMI was found among women with normotensive pregnancies. We have now added this to the discussion that now reads;

”If a larger sample size could be obtained in future studies, it would be of value to perform adjusted models correcting for potential confounders such as BMI, maternal age and parity.”

Page 11, lines 344-346

  • Do you think that the protein total in CSF from PE and E may be higher than normotensive. Do you measured the protein total level?

We did not measure the total protein level in our CSF samples due to restricted volumes. It would have been of interest to examine this in future studies. Since all the proteins measured were found in higher quantities it would be likely that the total protein level would be increased in the CSF of women with eclampsia and preeclampsia.

Reviewer 5 Report

Preeclampsia and Eclampsia are hypertensive disorders of pregnancy that cause significantly maternal morbidity and mortality. The evidence of neurological inflammation and blood-brain barrier dysfunction is lacking in human studies, although animal studies demonstrate the dysfunction. The study describes characterization of neuroinflammation and markers of blood-brain barrier dysfunction in patient samples from the PROVE biobank. 

Abstract:

1. Line 37: Please replace the word pathology. The authors did no histological examination of tissues, thus pathology is not an appropriate description. 

Introduction:

1. Line 54: Please be specify where the interleukins were measured in the study being described. There are many studies demonstrating increased interleukins in the plasma and placental samples of women with PreE. 

2. Line 56: No studies have assessed what in women in eclampsia?

Materials and Methods:

  1. Please include inclusion criteria for the study.
  2. Where the eclampsia patients initially diagnosed with preeclampsia?
  3. How can the authors make comparisons with collection of blood at so many different time points? This is a serious experimental design flaw. 

Results:

  1. Table 1 Magnesium sulfate treatment for preeclampsia patients is inaccurate. Should be either 3(75) or 4(100).
  2. Line. 190-191- How can this patients be excluded and the authors still report a of 4 for this group? was this patients excluded for all markers measured

Discussion

  1. Define abbreviations at their first mention: A1M, AMBP, Hb
  2. There is no discussion on the roles of these inflammatory markers in causing cerebrovascular pathology or disease. Their relevance to higher risk later in life or the mechanisms by which they contribute to dysfunction are not discussed. 
  3. Line 306- The authors report that when they restrict their analysis to matching eclampsia and preeclampsia, significant differences are still observed, but data not shown. This data needs to be shown. 

Author Response

Reviewer #5

Preeclampsia and Eclampsia are hypertensive disorders of pregnancy that cause significantly maternal morbidity and mortality. The evidence of neurological inflammation and blood-brain barrier dysfunction is lacking in human studies, although animal studies demonstrate the dysfunction. The study describes characterization of neuroinflammation and markers of blood-brain barrier dysfunction in patient samples from the PROVE biobank.

Abstract:

  • Line 37: Please replace the word pathology. The authors did no histological examination of tissues, thus pathology is not an appropriate description.

Thank you for this comment. Pathology is now replaced and the sentence reads;

“In conclusion, women with preeclampsia and eclampsia show evidence of neuroinflammation and an injured blood-brain barrier. These findings are seen in particular among women with eclampsia.”

Page 1, lines 36-37

Introduction:

  • Line 54: Please specify where the interleukins were measured in the study being described. There are many studies demonstrating increased interleukins in the plasma and placental samples of women with PreE.

Thank you for this comment, please see the reply to Reviewer 2. We have added information about these studies and the section now reads;

“In these studies, animals with induced preeclampsia demonstrated increased number of activated microglial cells and an increased cerebrospinal fluid (CSF) concentration of cytokines. In addition, the CSF-plasma albumin ratio was increased in preeclampsia as a measurement of blood-brain barrier impairment.”

Page 2, lines 55-59

  • Line 56: No studies have assessed what in women in eclampsia?

We apologize if this was unclear. We have now edited the sentence and it now reads;

”To our knowledge, no studies have assessed neuroinflammation and the blood-brain barrier in women with eclampsia.”

Page 2, lines 61-62

Materials and Methods:

  • Please include inclusion criteria for the study.

Inclusion criteria can be found in the materials and methods section under study cohort and reads;

“For this study, we included women with singleton pregnancies. Exclusion criteria were known neurological or cardiac disease. For normotensive women, additional exclusion criteria were chronic hypertension and diabetes mellitus. Preeclampsia and eclampsia were defined according to the American College of Obstetricians and Gynecologists (ACOG) 2020 Practice Bulletin. A woman was considered normotensive if she had no documented systolic blood pressure greater than or equal to 140 mmHg or a diastolic blood pressure greater than or equal to 90 mmHg during her pregnancy until discharge postpartum.”

Page 2-3, lines 81-89

  • Where the eclampsia patients initially diagnosed with preeclampsia?

In this sub-population of the PROVE biobank, all four women with eclampsia presented with generalized tonic-clonic seizures. None of them had a diagnosis of preeclampsia before the first fit. Though the definition of eclampsia in the PROVE biobank requires hypertension and proteinuria in addition to generalized tonic-clonic seizures and also absence of other underlying causes to seizures.

  • How can the authors make comparisons with collection of blood at so many different time points? This is a serious experimental design flaw.

Thank you for this comment. Please see our reply to Reviewer 1 and also pasted below;

We thank the reviewer for this important aspect and as you can see, we have also contemplated on the importance of this. For this study, due to the acute situations when women are delivered with preeclampsia and eclampsia, we were not always able to obtain the blood samples at the same moment as the cerebrospinal fluid. We have now added information on this in the result section and it now reads;

“When restricting the analyses by matching eclampsia and normotensive controls for day of plasma sampling (n=4 in each group), plasma-albumin ratio in women with eclampsia remained increased compared to controls (median, range, 10.4; 8.2-23.2 vs 3.3, 2.8-4.1, p=0.029).”

Page 8, lines 220-223

In addition, we removed “data not shown” on page 10 in the discussion.

 Results:

  • Table 1 Magnesium sulfate treatment for preeclampsia patients is inaccurate. Should be either 3(75) or 4(100).

We thank the reviewer for noticing this and have now corrected to 3(75).

  • 190-191- How can this patients be excluded and the authors still report a n of 4 for this group? was this patients excluded for all markers measured

This woman was included in all analyses. When performing the analyses with this individual excluded, results remained similar, this data was not included in the manuscript. This is now clarified and now reads;

Excluding the results of this participant did not significantly change the findings (data not shown).

Page 8, lines 212-213

Discussion

  • Define abbreviations at their first mention: A1M, AMBP, Hb

We thank the reviewer for this point and this is now written out.

  • There is no discussion on the roles of these inflammatory markers in causing cerebrovascular pathology or disease. Their relevance to higher risk later in life or the mechanisms by which they contribute to dysfunction are not discussed.

We have added a section about this to the discussion and it now reads;

“The presence of an injured blood-brain barrier and neuroinflammation in other medical conditions such as traumatic brain injury have been suggested as underlying factors of long-term neurological outcome such as cognitive dysfunction and chronic traumatic encephalopathy.(2, 3) There are similar hypotheses in the field of neurodegenerative disease with an initial insult to the blood-brain barrier and subsequent neuroinflammation as an underlying trigger to the development of later neurodegenerative disease.(4)

Treatments that restore or protect blood-brain barrier integrity and decrease neuroinflammation could be important to prevent and treat eclampsia and may possibly protect women from long-term cerebral adverse outcomes associated with preeclampsia such as cognitive decline, epilepsy and dementia.(5-7)”

Page 10, lines 308-318

  • Line 306- The authors report that when they restrict their analysis to matching eclampsia and preeclampsia, significant differences are still observed, but data not shown. This data needs to be shown.

Thank you for raising this point. Please see the reply to the question from Reviewer 1, also pasted below;

We thank the reviewer for this important aspect and as you can see, we have also contemplated on the importance of this. For this study, due to the acute situations when women are delivered with preeclampsia and eclampsia, we were not always able to obtain the blood samples at the same moment as the cerebrospinal fluid. We have now added information on this in the result section and it now reads;

“When restricting the analyses by matching eclampsia and normotensive controls for day of plasma sampling (n=4 in each group), plasma-albumin ratio in women with eclampsia remained increased compared to controls (median, range, 10.4; 8.2-23.2 vs 3.3, 2.8-4.1, p=0.029).”

Page 8, lines 220-223

In addition, we removed “data not shown” on page 10 in the discussion.

Round 2

Reviewer 1 Report

The following answer should be stated in the limitation of discussion section.  

  • The exact time of magnesium sulphate use should be stated. Magnesium sulphate might influence inflammatory factors and albumin.

This is also an important comment. To our knowledge, it is not yet known how magnesium sulphate affects the content of neuroinflammatory markers in the cerebrospinal fluid in humans whereas it has been shown to reduce neuroinflammatory activity through reduced expression of active neuroglial cells in animal models of preeclampsia. In this study, all women with eclampsia and preeclampsia had received magnesium sulphate before delivery whereas none in the normotensive groups did. If anything, this would have reduced the difference in neuroinflammatory markers between the group, should magnesium sulphate had lowered the neuroinflammatory activity in women with preeclampsia and eclampsia. In our study, magnesium sulfate was administered between 30-1 hours before delivery. These details have now been added to Table 1.

Reviewer 2 Report

The authors have provided a revised version of the manuscript. They properly answered all the questions and suggestions to improve it.